# Microalgae Cultivated under Magnetic Field Action: Insights of an Environmentally Sustainable Approach

Kricelle Mosquera Deamici [1], Katarzyna Dziergowska [2,3], Pedro Garcia Pereira Silva [4], Izabela Michalak [2], Lucielen Oliveira Santos [4], Jerzy Detyna [3], Sunita Kataria [5], Marian Brestic [6], Mohammad Sarraf [7,8,*] and Monirul Islam [9,10,*]

1 GreenCoLab—Associação Oceano Verde, Algarve University, 8005-139 Faro, Portugal
2 Department of Advanced Material Technologies, Faculty of Chemistry, Wrocław University of Science and Technology, Smoluchowskiego 25, 50-372 Wrocław, Poland
3 Department of Mechanics, Materials Science and Engineering, Faculty of Mechanical Engineering, Wrocław University of Science and Technology, Smoluchowskiego 25, 50-370 Wrocław, Poland
4 School of Chemistry and Food, Federal University of Rio Grande, Rio Grande 96203-900, Brazil
5 School of Biochemistry, Devi Ahilya Vishwavidyalaya, Indore 452001, India
6 Institute of Plant and Environmental Sciences, Faculty of Agrobiology and Food Resources, Slovak University of Agriculture, 94976 Nitra, Slovakia
7 Department of Horticulture Sciences, Faculty of Agriculture, Shahid Chamran University of Ahvaz, Ahvaz 61357, Iran
8 Laboratory of Chemistry, R & D Center, Niroo Gostar Lian Inspection Co., Bushehr 75177, Iran
9 Department of Sustainable Crop Production, Università Cattolica Del Sacro Cuore, Via Emilia Parmense 84, 29122 Piacenza, Italy
10 Department of Biochemistry and Molecular Biology, University of Massachusetts Amherst, Amherst, MA 01003, USA
* Correspondence: sarraf.science@gmail.com (M.S.); monirulislam@umass.edu or monirul.islam@unicatt.it (M.I.)

**Abstract:** Microalgae and cyanobacteria include procaryotic and eucaryotic photosynthetic microorganisms that produce biomass rich in biomolecules with a high value. Some examples of these biomolecules are proteins, lipids, carbohydrates, pigments, antioxidants, and vitamins. Currently, microalgae are also considered a good source of biofuel feedstock. The microalga-based biorefinery approach should be used to promote the sustainability of biomass generation since microalga biomass production can be performed and integrated into a circular bioeconomy structure. To include an environmentally sustainable approach with microalga cultures, it is necessary to develop alternative ways to produce biomass at a low cost, reducing pollution and improving biomass development. Different strategies are being used to achieve more productivity in cultivation, such as magnets in cultures. Magnetic forces can alter microalga metabolism, and this field of study is promising and innovative, yet remains an unexplored area. This review presents the current trends in the magnetic biostimulation of microalgae for the application of cultivated biomass in different areas of biotechnology, biofuel, and bioenergy production, as well as environmental protection.

**Keywords:** microalgae; magnetic field; growth rate; chemical composition; algal biorefinery; environmental safety; sustainability

## 1. Introduction

The microalga biorefinery concept assumes the conversion of biomass into marketable chemicals, such as biofuels as other high-value co-products [1]. Microalga biomass is an interesting feedstock for biorefineries due to its chemical composition, fast growth, ability to grow in low-quality water and, additionally, remove contaminants from wastewater, as well as capture and recycle carbon dioxide ($CO_2$) from industrial flue emissions [2]. The impacts to the environment caused by greenhouse gas emissions are improving the forces to

generate energy from renewable sources. Among the green raw materials that can be used to create biofuels, microalgae are considered an alternative to replace fossil fuels by playing an important role in global environmental issues, leading to a sustainable path to obtaining biofuels [3]. In order to further enhance alga growth and synthesis of biologically active compounds of commercial interest, electromagnetic biostimulation of living cultures has been proposed [2]. Magnetic fields (MFs), used as a physical treatment, have been gaining more popularity in microalga cultivation, since they are non-toxic and non-polluting, without secondary contamination. Furthermore, no external energy is required for MF treatment. As a result, this saves energy and protects the environment [4–7].

Exposure of algae to the MF action increases not only biomass production, but also its composition—the content of carbohydrates, essential amino acids, lipids, pigments (e.g., phycocyanin), and antioxidants, which guarantees interest in the food, cosmetics, and feed industries, e.g., [8–17]. MF-stimulated microalga cultivation with enhanced lipid content may be beneficial in biodiesel production; however, its manufacturing costs are still significantly higher than fossil diesel, e.g., [5,10,18]. Besides biodiesel, microalgae can constitute the raw material for bioethanol production via fermentation processes, biogas generated during anaerobic decomposition of biomass, or hydrogen produced by photobiological processes [19].

The use of wastewater, rich in phosphorus and nitrogen, as the growth medium for microalgae grown under MF exposure seems to be justified and beneficial for the environment [5,20]. Microalgae treated with MFs may eliminate from wastewater not only inorganic, but also organic pollutants—e.g., starch [20] or dyes [21]. In this process, after wastewater treatment, biomass may be used for biofuel production, e.g., [4,5]. Exposure to MFs may also increase $CO_2$ biofixation by microalgae, since they are able to use atmospheric $CO_2$ (from industrial flue gases) as a carbon source to produce valuable compounds, while reducing the negative impact on the environment of this greenhouse gas [17].

In order to achieve satisfactory results in biorefineries using microalga biomass exposed to MFs as a feedstock, the optimization of this process is necessary, as well as conducting detailed research in real conditions, not only indoors cultivation, but also outdoors in open-raceway ponds. The biological consequences of magnetic exposure in microalgae are dependent on the magnetic intensity, frequency, and exposure period [6,9].

The aim of this review is to present the current trends in electromagnetic biostimulation of microalgae for the application of biomass in different areas of biotechnology, biofuel, and bioenergy production, as well as environmental protection. Although this field has emerged as innovative, it remains an unexplored area, especially concerning outdoors MF applications. The investigation of different microalga species, as well as MF parameters such as exposure time and intensity, combined with the application period during cultivation and different devices used to apply MFs to achieve satisfactory outcomes is in progress. This review is organized as follows: Section 2 presents the link between microalga cultures and magnetic fields and the related bio-effects; Section 3 explains the algae-based biorefinery concept; Section 4 highlights the sustainable and environmentally friendly application of microalgae; Section 5 outlines the final considerations.

## 2. Microalga Cultures and Magnetic Fields

Microalgae and cyanobacteria (blue algae) are photosynthetic microorganisms, being found in different ecosystems (aquatic and terrestrial), which generate biomass using different nutrient sources, $CO_2$, and illuminance. These microorganisms represent a wide variety of species that may survive in an extensive variety of environmental conditions [22,23]. In general, these microorganisms develop with a relatively fast growth rate and simple nutritional requirements. Furthermore, they may be phylogenetically classified as prokaryotic or eukaryotic and are considered favorable for biomass and different biomolecules' production, such as proteins, carbohydrates, and lipids [24,25].

In this context, microalgae may produce biomass by three specific systems: photoautotrophic, heterotrophic, and mixotrophic cultivation [26]. Photoautotrophic cultivation

is the most-common system to produce biomass and uses solar illuminance and $CO_2$ as an energy source. Thus, it is important to highlight the high photosynthetic efficiency of these microorganisms, capable of capturing $CO_2$ from the environment and using it for their growth [27,28].

In mixotrophic cultures (a variant of heterotrophic system), organic compounds and $CO_2$ are assimilated by photosynthetic and respiratory metabolism to develop a rapid growth rate and biomass productivity. In this system, organic carbon sources are added, such as simple or complex sugars and glycerol. The ability to easily assimilate available carbon in the medium may promote the accumulation of some specific macromolecules, such as lipids and carbohydrates [27,29].

The production of metabolites of interest by these microorganisms is determined by different widely studied key factors, such as nutrient composition, illuminance, temperature, pH, $CO_2$ concentration, and agitation/aeration [22,30,31]. Thus, the microalgal biomass produced may be destined for different applications (Figure 1), including for use in industrial processes [14,27], but this also needs to be highlighted as a potential resource for biofuel production [23,31].

**Figure 1.** Factors influencing algae cultures and possibilities for industrial applications of microalga biomass.

Several technological approaches have been used in microalga cultures to increase biomass yield [27,32]. However, new strategies have been studied to induce physical or chemical stress during cultivation. They can be applied individually or synergistically, activating cellular defense mechanism by these microorganisms, mainly stimulating carbohydrate and lipid production, essential macromolecules for the use in biorefineries [23].

Regarding these new strategies that have been employed in microalga cultivation, MF application has been shown to be an alternative technological approach, due to its interaction with biological systems, positively influencing the production of biomass and compounds of interest [6,15].

This technological approach has been used in bioprocesses interacting with cell microorganisms to improve some compounds of interest, such as biomass and ethanol by *Saccharomyces cerevisiae* [33,34], glutathione by *S. cerevisiae* [35], biomass and carotenoids by *Phaffia rhodozyma* [36], laccase by *Candida tropicalis* [37], red and yellow pigment by *Monascus purpureus* [38], citric acid and cellulase by *Aspergillus niger* [39], inulinase production by *Geotrichum candidum* [40], lipid and pigments by *Chlorella kessleri* [8], protein and phycocyanin by *Spirulina* sp. LEB 18 [13], carbohydrate by *Chlorella minutissima* [41], and biomass and lipid content by *Chlorella homosphaera* [10].

## 2.1. Bioeffects of Magnetic Field on Microalga Cultivation

MFs are considered an environmental factor by which magnetic forces in space interact with all living systems during the evolutionary process. Thus, MFs produced by humans have become a growing part of the biosystem, in which living organisms need to adapt to this physical factor, because it directly influences some of their biological functions [42]. Although MF application in bioprocesses is considered a new, non-toxic, low-cost, and easy-to-implement technology, its mechanisms of action in biological systems have not yet been fully elucidated. Responses about its effects are non-linear, varying with the parameters employed in cultivation [11,31].

The effect of MF application in microalga cultures has been investigated in the last decade to understand its effects on the metabolism of these cells. Their responses may be considered null, negative, or positive, depending on changes caused in the biological system. Furthermore, factors, such as microalga species, exposure time, and MF intensity, may increase biomass production of high value-added products [26,41], as shown in Table 1.

**Table 1.** Positive effect of MF application on microalga cultures.

| Microalga Specie | Type of Microalgae | MF Intensity and Exposure Time | Cultivation Time (Days) | Positive Effect | Reference |
|---|---|---|---|---|---|
| *Scenedesmus obliquus* | Green algae | 50 mT for 1 h/d | 6 | 11.5% chlorophyll *a* content | [4] |
| *Spirulina* sp. | Blue-green algae | 60 mT for 24 h/d | 15 | 57.2% phycocyanin content | [13] |
| *Chlorella kessleri* | Green algae | 30 mT for 24 h/d | 10 | 25% carotenoid content | [8] |
| *Chlorella vulgaris* | Green algae | 30 mT for 1 h/d | 7 | 84% protein content | [43] |
| *Spirulina* sp. | Blue-green algae | 60 mT for 24 h/d | 15 | 16.7% protein content | [13] |
| *Tribonema* sp. | Yellow-green algae | 30 mT for 24 h/d | 9 | 85.4% protein content | [20] |
| *Chlorella kessleri* | Green algae | 10 mT for 24 h/d | 12 | 47% lipid content | [12] |
| *Chlorella kessleri* | Green algae | 30 mT for 24 h/d | 10 | 13.7% lipid content | [8] |
| *Haematococcus pluvialis* | Red algae | 16.9 mT for 24 h/d | 7 | 25% lipid content | [18] |
| *Chlorella homosphaera* | Green algae | 15 mT for 1 h/d | 15 | 22.4% lipid content | [10] |
| *Spirulina* sp. | Blue-green algae | 30 mT for 24 h/d | 15 | 45.5% lipid content | [44] |
| *Chlorella kessleri* | Green algae | 10 mT for 24 h/d | 12 | 8.5% carbohydrate content | [12] |
| *Chlorella fusca* | Green algae | 60 mT for 24 h/d | 15 | 24.6% carbohydrate content | [45] |
| *Chlorella minutissima* | Green algae | 30 mT for 24 h/d | 12 | 162.9% carbohydrate content | [41] |
| *Chlorella fusca* LEB 111 | Green algae | 25 mT for 1 h/d | 15 | 10.6% carbohydrate content | [15] |

Microalga cells are composed of a complex metabolic system [6]. Thus, MF application may biochemically influence the activation of specific enzyme systems and metabolic pathways and alter plasma membrane flux and gene transcription (Figure 2), due to the cell defense mechanisms caused by environmental stresses [20,46].

In this context, Albuquerque et al. [47] and Zhang et al. [42] reported that the regulation of cellular metabolism by gene transcription changes the plasma membrane flux, facilitating the entry of $Ca^{2+}$ into the cytoplasm (responsible for cell growth). Activation of the metabolic and enzymatic systems to produce macromolecules of interest or secondary metabolites is considered to have positive effects caused by exposure to MFs in microalga cultures.

Biomolecules of high-added value produced by microalgae are not induced by a specific factor. Thus, the use of factors combined with MF application may influence the production of specific compounds, such as carbohydrates or lipids. However, the responses related to MF application in microalga cultivation need to be further explored at the molecular level to better understand its mechanism of action [31,41].

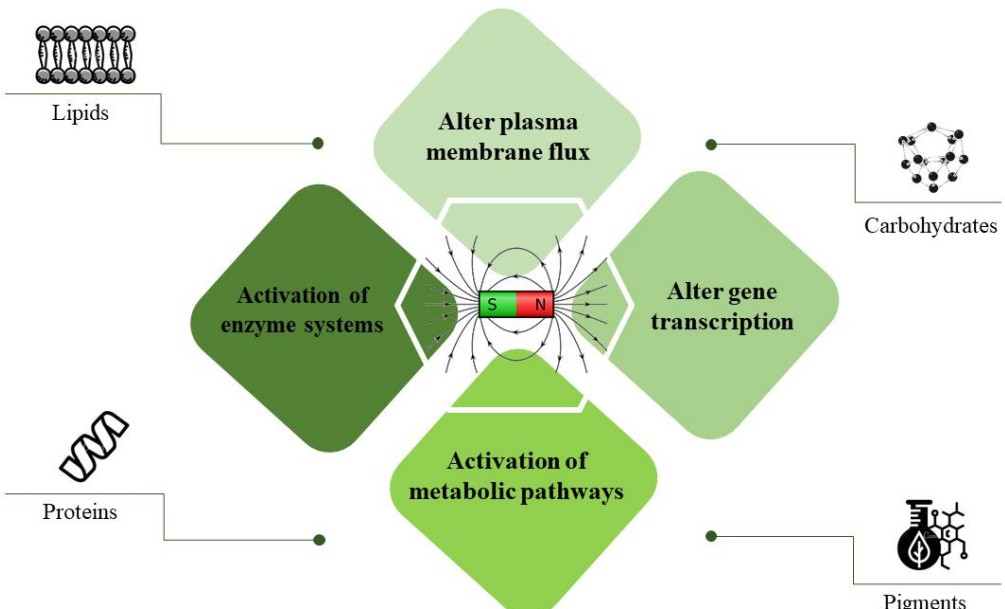

**Figure 2.** Different magnetic field actions on microalga cells.

### 2.2. Magnetic Field Effect on Microalga Cell Growth and Cell Size

An interesting approach to microalga exposure to MFs is the assessment of their effects on the specific growth rate. Several publications have established that the highest biomass concentration is associated with the maximum growth rate [12,48,49]. The comparison of the cell growth rate and biomass increase for different microalga species exposed to MFs with different strengths and exposure times is shown Table 2.

The influence of MFs on the growth rate and biomass production has been most often examined in cyanobacteria *Spirulina* sp. and microalga *Chlorella* sp. This may be because commercial-scale cultivation of these species has already been introduced for various uses. *Spirulina* sp. is claimed to be a potential source of antioxidants, pigments, carbohydrates [17,54], and proteins [14,54] and, therefore, may be used in the food industry [14,54] and in the biofuel sector [55]. *Chlorella* sp. is known for promoting the immune and cardiovascular systems. Microalgal carotenoids, in turn, are some of the strongest natural antioxidants with high biomedical value [12]. *Chlorella* sp. contains proteins, vitamins, liposoluble compounds, glycolipids [14], and fatty acids and may be used for biodiesel production or food enrichment [8,45].

Table 2 indicates that MFs with an intensity of 10 to 500 mT have been used to biostimulate microalga growth, but the most common values were up to 100 mT. The highest biomass, by comparison with the control group, was noted at 60 mT for *Spirulina* sp. (increase by 95%), at 10 mT for *Dunaliella salina* (increase by 84%), and at 10 mT for *Chlorella kessleri* (increase by 77%).

Interesting results were described by Yamaoka et al. [51], who found that the composition of the medium may impact microalga cells' response to SMFs. This stimulation in conjunction with Fe-EDTA (10 mg $L^{-1}$) had a negative effect on *Dunaliella salina* biomass concentration by comparison with a microalga cultivated under the influence of SMFs and a medium without the addition of Fe-EDTA. In turn, when this microalga was grown in a medium containing 1 mg $L^{-1}$ of Fe-EDTA and exposed to SMF stimulation, the biomass concentration increased up to about 84% by comparison with the group without any SMF stimulation. In the case of cells cultured with the addition of Fe-EDTA, a magnetic effect was observed with 10 mT and decreased with a higher intensity of the SMF (20, 50, 100, and 200 mT). These results clearly indicated that the components of the culture medium may significantly affect the stimulation of cell growth.



**Table 2.** Effect of the magnetic field exposure on the microalga growth rate and production by comparison with the control group.

| Microalga Species | Intensity/Exposure Time | Growth Rate Increase (↑)/Decrease (↓) | Biomass Increase (↑)/Decrease (↓) | Reference |
|---|---|---|---|---|
| *Chlorella fusca* | 60 mT/24 h/d | ↓ 12% | ↑ 21% | [45] |
| *Chlorella fusca* | 60 mT/1 h/d | ↓ 14% | ↑ 13% | [17] |
| *Chlorella fusca* | 25 mT/24 h/d (outdoors) 1 h/d (indoors) | ↓ 18%—outdoors ↑ 100%—indoors | ↑ 31%—outdoors ↑ 19%—indoors | [16] |
| *Chlorella homosphaera* | 30 mT/1 h/d | 106% | ↑ 21% | [10] |
| *Chlorella kessleri* | 10 mT/ every 18 min | ↑ 126%—in flask ↑ 88%—in raceway pond | ↑ 77% | [12] |
| *Chlorella pyrenoidosa* | 500 mT/3 h/d | n.a. | ↑ 12% | [5] |
| *Chlorella vulgaris* | 10 mT/12 h | ↑ 72% | n.a. | [47] |
| *Chlorella vulgaris* | PEMF * 0.07 mT, 1 Hz/4 h/d | n.a. | ↑ 315% compared to 0.09 mT | [50] |
| *Dunaliella salina* | 10 mT, with 1 mg/L Fe-EDTA | n.a. | ↑ 84% | [51] |
| *Nannochloropsis oculata* | 10 mT/n.a. | n.a. | ↑ 43% | [52] |
| *Spirulina platensis* | 10 mT/n.a. | n.a. | ↑ 10% | [48] |
| *Spirulina platensis* | 250 mT/n.a. | n.a. | ↑ 47% | [53] |
| *Spirulina platensis* | 30 mT/ 3, 6 and 12 h/d | n.a. | ↑ 16% (the highest for 6 h/d) | [54] |
| *Spirulina platensis* | 30 mT/1 h/d | n.a. | ↑ 49% | [55] |
| *Spirulina* sp. | 60 mT/1 h/d | ↑ 48% | ↑ 95% | [13] |
| *Spirulina* sp. | 25 mT/24 h/d | the same for outdoors and indoors as in control | ↑ 6%—outdoors ↓ 6%—indoors | [11] |
| *Spirulina* sp. | 60 mT, 1.9 g/L NaNO$_3$ 24 h/d | ↑ 17% | ↑ 37% | [14] |

* PEMF—pulsed electromagnetic field, n.a.—not available.

Hirano et al. [48] evaluated the responses of *Spirulina platensis* (growth rate, cell concentration, glyceroglycolipid, sugars, and pigments such as chlorophyll, β-carotene, and phycocyanin) cultivated under heterotrophic and autotrophic conditions with SMF induction. For autotrophic cultures, the growth rate and biomass concentration increased when 10 mT was applied, as well as the sugar and phycocyanin content. For heterotrophic cultures, the magnetic effect was null.

In the literature, there are also some studies that tried to link the microalga growth rate with the cell size [56]. This relation is not always applicable, because cell size is not always an indication of microalga growth [57]. However, cell size may be evidence the rate of cell division [12]. Small et al. [12] observed a reduction in the size of *Chlorella kessleri*, with a high biomass production and growth rate, when the microalga was exposed to 10 mT by comparison with the control group. This indicates that smaller cells undergo faster division [12]. In the study of Oliveira [52], *Nannochloropsis oculata* exposed to 5 mT had the largest cell size, which theoretically could imply slower growth.

### 2.3. Magnetic Field Effect on the Chemical Composition of Microalgae

The key issue in MF application to microalga cultivation is the assessment of its impact on the biomass composition. In the case of biologically active compounds, such as proteins, carbohydrates, lipids, and pigments, exposure of microalgae to SMFs may increase their synthesis. Table 3 shows examples of the effects of MFs on the biochemical composition of different microalga species.

**Table 3.** The influence of the magnetic field on the chemical composition of microalgae. The best condition is highlighted in bold.

| Microalga Species | Intensity/Exposure Time | Effect of MF vs. Control | Reference |
|---|---|---|---|
| Carbohydrates | | | |
| *Chlorella fusca* | 30, **60** mT/1, **24** h/d | 31 vs. 25% *w/w* | [45] |
| *Chlorella fusca* | 30, **60** mT/1, **24** h/d | 25 vs. 17% *w/w* | [17] |
| *Chlorella fusca* | 25 mT/**1**, 24 h/d | 34 vs. 31% *w/w*—outdoors<br>28 vs. 25% *w/w*—indoors | [15] |
| *Chlorella homosphaera* | 15 mT/1 h/d<br>30, **60** mT/1, **24** h/d | 45 vs. 47% *w/w* | [10] |
| *Chlorella kessleri* | 10 mT/every 18 min | 42 vs 39% *w/w* | [12] |
| *Chlorella kessleri* | **30**, 60 mT/1, **24** h/d | 21 vs. 19% *w/w* | [8] |
| *Spirulina platensis* | 30 mT/**3**, 6, 12 h/d | 13 vs. 10% c.d.w. | [54] |
| *Spirulina platensis* | 30 mT/1, **24** h/d | 16 vs. 13% *w/w* | [55] |
| *Spirulina* sp. | **30**, 60 mT/1, **24** h/d | 30 vs. 13% *w/w* | [13] |
| *Spirulina* sp. | 30, **60** mT/24 h/d | 7–15 vs. 12–22% *w/w* | [14] |
| Pigments: Chlorophyll *a* | | | |
| *Chlorella kessleri* | 10 mT/every 18 min | 2.9 vs. 2.5% *w/w* | [12] |
| *Spirulina platensis* | 250 mT/n.a. | 13.5 vs. 10.3 g/L | [53] |
| *Spirulina platensis* | 30 mT/**1**, 24 h/d | 2.6 vs. 2.2% *w/w* | [55] |
| *Nannochloropsis oculata* | 5, 10, 15 mT/n.a. | 1.9 to 2.0-fold increase at 5 and 10 mT, respectively; no change at 15 mT | [52] |
| Pigments: Chlorophyll *b* | | | |
| *Chlorella kessleri* | 10 mT/every 18 min | 0.87 vs. 0.53% *w/w* | [12] |
| Pigments: Carotenoids | | | |
| *Chlorella kessleri* | 10 mT /every 18 min | 0.08 vs. 0.07% *w/w* | [12] |
| Pigments: Phycocyanin | | | |
| *Spirulina platensis* | 30 mT/**1**, 24 h/d | 11.5 vs. 12.6% *w/w* | [45] |
| *Spirulina sp.* LEB 18 | 30, **60** mT/**1**, 24 h/d | 13.2 vs. 8.4 g/L | [13] |
| *Spirulina platensis* | 10 mT/n.a. | increase by 54% vs. control | [48] |
| Protein | | | |
| *Chlorella fusca* | **30**, 60 mT/**1**, 24 h/d | 64 vs. 60% *w/w* | [45] |
| *Chlorella fusca* | **30**, 60 mT/**1**, 24 h/d | 56 vs. 52% *w/w* | [17] |

**Table 3.** *Cont.*

| Microalga Species | Intensity/Exposure Time | Effect of MF vs. Control | Reference |
|---|---|---|---|
| *Chlorella fusca* | 25 mT/**1**, 24 h/d | 33 vs. 29% *w/w*—outdoors<br>36 vs. 28% *w/w*—indoors | [15] |
| *Chlorella homosphaera* | **15** mT/**1** h/d<br>30, 60 mT/1, 24 h/d | 23 vs. 18% *w/w* | [10] |
| *Chlorella kessleri* | 10 mT/every 18 min | 32 vs. 30% *w/w* | [12] |
| *Chlorella kessleri* | **30**, 60 mT/**1**, 24 h/d | 59 vs. 54% *w/w* | [8] |
| *Spirulina platensis* | 30 mT/3, **6**, 12 h/d | 65 vs. 59% c.d.w. | [54] |
| *Spirulina platensis* | 30 mT/**1**, 24 h/d | 67 vs. 67% *w/w* | [55] |
| *Spirulina* sp. | 30, **60** mT/1, **24** h/d | 73 vs. 63% *w/w* | [13] |
| *Spirulina* sp. | 30, **60** mT/24 h/d | 66–73 vs. 58–71% *w/w* | [14] |
| Lipids | | | |
| *Chlorella fusca* | **30**, 60 mT/1, **24** h/d | 13 vs. 13% *w/w* | [45] |
| *Chlorella fusca* | **30**, 60 mT/**1**, 24 h/d | 19 vs. 18% *w/w* | [17] |
| *Chlorella fusca* | 25 mT/**1**, 24 h/d | 32 vs. 30% *w/w*—outdoors<br>36 vs. 34% *w/w*—indoors | [15] |
| *Chlorella homosphaera* | 15 mT/1 h/d<br>**30**, 60 mT/**1**, 24 h/d | 44 vs. 29% *w/w* | [10] |
| *Chlorella kessleri* | 10 mT/every 18 min | 20 vs. 25% *w/w* | [12] |
| *Chlorella kessleri* | 30, **60** mT/**1**, 24 h/d | 24 vs. 21% *w/w* | [8] |
| *Chlorella pyrenoidosa* | 500 mT/3 h/d | 216 vs. 212 mg/g c.d.w. | [5] |
| *Chlorella vulgaris* | PEMF * **0.06**, 0.07, 0.08, 0.09 mT at 1 Hz/4 h/d | 55 d.w. vs. 49% d.w. | [50] |
| *Spirulina platensis* | 30 mT/3, 6, **12** h/d | 18 vs. 16% c.d.w. | [54] |
| *Spirulina* sp. | 30, **60** mT/1, **24** h/d | 10 vs. 8% *w/w* | [13] |
| *Spirulina* sp. | 30, **60** mT/24 h/d | 9–15 vs. 11–17% *w/w* | [14] |

c.d.w.—cell dry weight, n.a.—not available, in bold—the best experimental conditions. * PEMF—pulsed electromagnetic field.

The highest variability in the compounds produced, depending on the MF intensity and exposure time, was observed for carbohydrates, e.g., [12,45,48]. However, there is no clear trend in the stimulation of biomolecules' production by microalgae under the influence of MFs, and most likely, this may be only species-dependent. For the strains of *Chlorella* sp. and *Spirulina* sp. evaluated, an increase in sugar content was observed for MFs from 30 to 60 mT and a longer exposure time—24 h/d (Table 2). Stimulation of *Chlorella* sp. with MFs rather than *Spirulina* sp. is recommended to produce biomass with a higher content of the substrate. According to Deamici et al. [45], *Chlorella fusca* exposed to a static MF may serve as a feedstock for the production of third-generation bioethanol due to the increased carbohydrate content in this microalga.

Some literature data indicate a decrease in carbohydrate content in microalga biomass cultivated with MFs. Hirano et al. [48] found that the carbohydrate content in *S. platensis* was reduced by 0.6-fold when cells were subject to 70 mT by comparison with the assay where no MF was applied. The change in carbohydrate content was associated with a lower total pigment content (chlorophyll *a* and phycocyanin) and a 0.5-fold growth rate, which indicated a reduction in the photosynthetic capacity of the cells [48]. Oliveira [52] showed that for the highest intensity of applied SMF (15 mT), *N. oculata* cells exhibited a 23% reduction in carbohydrate content by comparison with the assay control. It is worth noting that the chlorophyll-*a* content ($0.30 \pm 0.04\%$) was comparable to the control ($0.37 \pm 0.09\%$)

and the carotenoid content increased 66.7% by comparison to the control [52]. Exposure of microalgae to a static MF may significantly influence the modification of monosaccharides' composition. The content of different substrates (glucuronic acid, arabinose, fucose, galactose, and rhamnose) increased, whereas glucose content was reduced in *Spirulina* sp. exposed to SMFs for 1 and 24 h/d [16].

Pigment content in microalgae appears to be positively influenced by SMF treatment, as shown in Table 3. The studied MF range was from 10 to 250 mT. Most authors reported increased content of chlorophyll *a* (primary pigment), accessory pigments (e.g., chlorophyll *b*), as well as carotenoids and phycocyanin in microalgae exposed to MFs. Hirano et al. [48] found that phycocyanin content in *Spirulina platensis* was the highest for 10 mT—148 mg/g of cell (54% higher by comparison with the control group—geomagnetic field 0.05 mT)—but decreased with increase in MF intensity from 20, 35, 40, to 70 mT (for 70 mT, it was by 20% lower than for the control group). Stimulation of chlorophyll synthesis may enhance light capture, which is essential for photosynthesis and may accelerate microalga growth. According to Young and Frank [58], pigments, such as carotenoids, due to their antioxidant properties, play an important role in cells—prevent oxidative stress.

The influence of SMFs on protein content in microalgae was examined mainly for a range of MF from 10 to 60 mT. Oliveira [52] presented that the protein level in *N. oculata* remained constant (about 25%) in relation to the differentiated impact of SMFs within the range of 5–15 mT. A more visible effect of MF exposure on protein content was noted with *Spirulina* sp. than with *Chlorella* sp., and with 1 h/d, the best results occurred (Table 3). Veiga et al. [14] showed that MFs applied to a *Spirulina* sp. culture had no effect on the protein content, but increased the protein digestibility and protein solubility and reduced the carbohydrate content by comparison with the control culture, indicating that this biomass may be used in the food industry as a component in the production of protein supplements.

The lipid content increase in microalga biomass under the influence of MFs is the most studied. However, the exposure of microalgae to MFs has the slightest effect on the lipid content among determined biomolecules. Most authors reported no changes in the lipid level in microalgae, but *Chlorella* sp. is a naturally richer source of lipids than *Spirulina* sp. (Table 3).

It is important to consider that MF application to biological systems may cause oxidative stress by the action of free radicals or the interaction with the membrane's structure [12,59,60]. Microalga cultivated under MF influence may use additional energy to counteract the negative effect of increased oxidative stress, which consequently, may reduce the level of biomolecules, such as carbohydrates and pigments. Small et al. [12] showed that the antioxidant activity of *Chlorella kessleri* cells cultivated with 10 mT decreased by 35% compared to the control group, as a result of increased oxidative stress. Oliveira [52] indicated that during exposure of *N. oculata* to MFs, the level of free radicals was not high enough to reduce the carotenoid content. Microalga exposed to 15 mT had a carotenoid content from 29 to 33% lower than for intensity values of 5 and 10 mT, respectively, but 67% higher than in the control group. Microalga *Nannochloropsis gaditana* exposed to MFs had higher superoxide dismutase and catalase activity, as well as synthesized significantly high amounts of antioxidant pigments (especially violaxanthin) as a part of the non-enzymatic defense system, as compared to the control culture [9]. To better understand the effect of MFs on cells, a more comprehensive summary of the action mechanism in different microalga species is elucidated in the work of Santos et al. [61].

## 3. Microalga-Based Biorefinery

The term biorefinery is well known as an industrial plant where crude oil is converted into useful oil products, such as petroleum, gasoline, and fuel oils, among others. The concept includes the process of obtaining energy and high-value products through biomass transformation in a sustainable way [3]. The term microalga-based biorefinery is similar, where microalga biomass is transformed into high-value-added bioproducts, such as

biofuels (biodiesel, bioethanol, biogas, biohydrogen), bioplastics, pigments, nutraceuticals, and biofertilizers [62,63]. The microalga-based biorefinery may be viable, since these microorganisms are a biofuel feedstock, besides being able to capture atmospheric $CO_2$ and produce biomass rich in many bioproducts, through wastewater bioremediation [64].

Microalga cultivation has many advantages that make this type of biorefinery feasible. These microorganisms may be cultivated under sunlight and atmospheric $CO_2$, if they can use it as a carbon source, minimizing cultivation costs. Besides, the microalga culture does not require arable land and dependence on seasonality. Microalgae grow photosynthetically, depending only on sunlight, $CO_2$, and nutrients from the culture medium.

The biorefinery concept was created to describe the biofuels and high-value biomolecules' production from biomass by the integration of bioprocessing with a low environmental impact at a sustainable cost. The microalga-based biorefinery fits this concept, since it is possible to produce biofuels from biomass rich in lipids and carbohydrates, in outdoors conditions with alternative culture media, which is a way to reduce production and energy costs. The biorefinery is a design for sustainable waste reuse and energy production. A microalga-based biorefinery is shown in Figure 3.

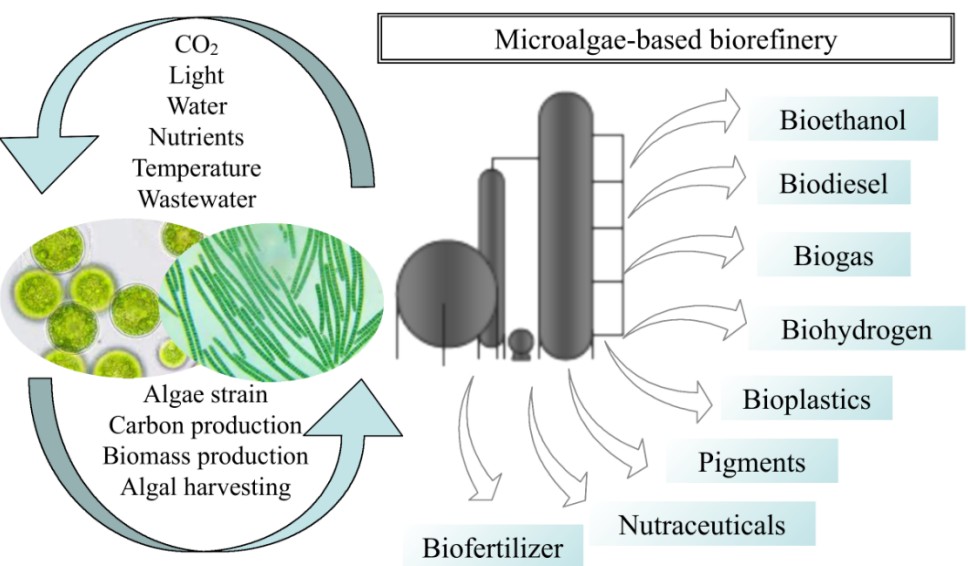

**Figure 3.** General scheme of microalga-based biorefinery.

Microalga biorefineries are developed with the aim to reduce biomolecules' production costs. For this purpose, the implementation of an integrated production process is necessary to achieve its sustainability, aiming at waste reduction and efficient transformation of biomass into energy, polymers, and food supplements. The integration of $CO_2$ mitigation and biofuel production, named $CO_2$-neutral fuels, has been used in the biorefinery approach. This environmental challenge is one of the most urgent in the world, given the huge gaseous emissions and their consequential climatic changes and warming effects [65].

High-value co-products and cultivation strategies must be used for a microalga-based biorefineries to be economically viable. The most-promising strategies involve microalga cultivation under atmospheric $CO_2$ mitigation, the use of wastewater as a substitute for culture media, and finally, the extraction of bioproducts from microalga biomass, as the feedstock for biofuel production [66]. Besides, MF strategies during microalga cultivation can be considered an important ally in the biorefinery approach, since this could enhance the biomass and compound production [11].

### 3.1. $CO_2$ Mitigation

$CO_2$ emissions from fossil fuels and industrial processes represented around the world 34 gigatons of $CO_2$ equivalent (Gt$CO_2$e) in 2020. Compared to the previous year, $CO_2$

emissions were reduced by 5.4% due to the COVID-19 pandemic and its impacts. However, an increase of 4.8% in 2021 has been estimated, contributing to global warming, which induces an increase in temperature and, consequently, impacts climate changes [67].

In this context, microalga cultures have been highlighted as important allies in $CO_2$ mitigation in the atmosphere by biological fixation. As photoautotrophic organisms, microalgae efficiently capture $CO_2$ by their photosynthetic system, being able to synthesize it as a carbon source for their development and convert it into $O_2$ [68]. In addition, microalga cultures have a high $CO_2$ fixation rate and, consequently, potential for renewable energy generation [69].

Thus, as an alternative to the problem of greenhouse gas emissions, microalga cultivation on an industrial scale has become a viable alternative to combat atmospheric pollution. In addition, due to the capacity for $CO_2$ biofixation, microalga cultures have some advantages, such as a high cell growth rate due to the consumption of $CO_2$ as the carbon source, easy harvesting, tolerance to high temperatures, the fixation of flue gases such as sulfur oxide ($SO_x$) and nitrogen oxide ($NO_x$), and the ability to be cultivated with wastewater from industrial treatment [70,71].

Microalga biomass is about 50% carbon, which is assimilated mainly by capturing $CO_2$ from the atmosphere. Thus, to produce 100 tons of microalga biomass, it is necessary to assimilate 183 tons of $CO_2$ [70]. To reduce $CO_2$ losses to the environment, technological approaches, such as pH control and MF application, assist the metabolism of $CO_2$ fixation by the microalgae during cultivation [17,72]. The study of Deamici et al. [17] evaluated *Chlorella fusca* cultivation in different conditions of MF application and $CO_2$ biofixation, which answered positively with regard to the MF's action, increasing the biofixation rate by 50% with 60 mT applied for 1 h d$^{-1}$. These authors demonstrated that, with MF application, it is possible to combine a higher biomass concentration with a high $CO_2$ fixation rate as a viable alternative. Deamici et al. [17] demonstrated that cultures under MF application reduced $CO_2$ as a greenhouse gas in the atmosphere, besides reducing the cost of microalga-based biomass production.

*3.2. Wastewater Treatment*

Wastewater is generated during domestic and industrial activities and is an environmental concern, since improper treatment and disposal represent a severe risk to the environment and the health of the communities nearby. Wastewater treatment is considered one of the main environmental problems of the 21st Century [73]. Wastewater content depends on the source; it is usually rich in organic and inorganic nutrients, with high biological and chemical oxygen demand (BOD and COD, respectively), which become a risk if untreated. Besides, many nutrients may be in industrial wastewater, such as nitrogen, phosphorus, and organic carbons, as well as pollutants, such as pharmaceutical compounds, dyes, and heavy metals [64,74].

Heterotrophic cultivation of microalgae allied with wastewater treatment has been identified to address this environmental issue. Microalgae are considered an efficient alternative for wastewater treatment by removing organic/inorganic loads and using them as a nutrient supplement (nitrogen and phosphorus) for their growth, besides providing useful biomass [75,76]. Wastewater treatment by these microorganisms represents a biocircular economy approach, since microalga growth in wastewaters provides the bioremediation of wastewaters and produces biomass.

The diversity of microalgae is huge and offers the possibility to use different conditions, depending on the microalga selected. Wastewaters usually contain high levels of constituents not ideal for optimal microalga growth, such as high $CO_2$ levels, high ammonia levels, and acidic/basic pH. Nevertheless, there are microalga species that are also resistant to these conditions, high/low temperature, and high salinity [77], among others, and their use in wastewater treatment is a promising alternative. As an example, in the literature, an extensive number of microalga species used in wastewater treatment have

been reported, such as *Chlorella* sp., *Scenedesmus* sp. [78], *Spirulina* sp. [79], *N. oculata* [80], and *Phormidium fragile* [81].

Wastewater treatment based on microalgae depends on strains capable of tolerating extreme conditions. Furthermore, the wastewater composition must be known, so that all processes can be adequately designed and operated [82]. The COD/BOD levels and nitrogen/phosphorus from agro-industrial wastewater are considered extreme for microalga cultivation. COD results in higher levels of water turbidity, affecting microalga growth by reducing light penetration. Another problem is the toxic natural ammonia content in the culture. Ammonia may affect the oxygen involved in photosystem II, alter photosynthesis, and interfere in the electron gradient in thylakoid membranes [83].

Otherwise, wastewater treatment with microalgae has many advantages, such as the feasibility of recycling the nutrients from the wastewater and their conversion into biomass, which may be used in other areas, reducing costs and applying the biocircular economy approach. On the other hand, the wastewater provides $CO_2$, which turns into a useful culture medium for these microorganisms, since the carbon, nitrogen, and phosphorus molecular ratio (106:16:1—C:N:P 1⁄4) in marine organic is retained, allowing great production rates [84,85]. Additionally, the microalgae's effects in wastewater are mitigated by removing nitrogen and carbon, contributing to biodiversity and helping to reduce eutrophication [86].

### 3.3. Different Ways of MF Application to Microalga Cultivation

There are considerable gaps in the research on growing microalgae to maximize biomass production. In addition, there is advanced research on the technological aspects of microalga cultivation using static and electrically generated MFs. Deamici et al. [13] evaluated the effect of MFs generated by permanent magnets (the most common use) and solenoids (SMFs) on *Spirulina* sp. LEB 18's growth and its chemical composition. MFs were applied around a vertical tubular photobioreactor, as demonstrated in Figure 4. Microalga growth under static MF application was superior (20.5%) by comparison to the control assay.

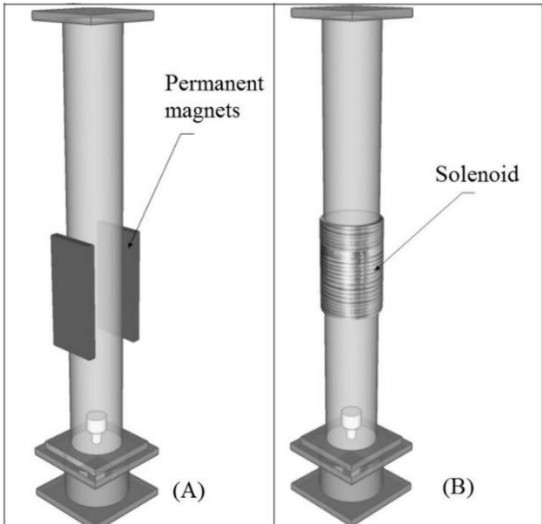

**Figure 4.** MF application by magnets (**A**) and with solenoids (**B**) in a vertical tubular photobioreactor. Source: [13].

The same group of authors [11] proposed applying six magnets (with an average intensity of 25 mT) in a raceway pond for 1 h/d (in the photoperiod of light) and throughout the culture for 24 h for 15 days. The effect of the MFs on *Spirulina* sp. was more pronounced when they were applied throughout the cultivation (24 h)—higher concentration of the biomass with increased biomolecule production.

Luo et al. [87] constructed a device for MF application (40, 80, and 150 mT for 2 h/d) on a *Chlorella vulgaris–Bacillus licheniformis* consortium used in sewage treatment. It consisted of a spiral regulating rod, a graduation plate, a movable magnetic pole, a fixed magnetic pole, and a container (Figure 5). As the intensity of the MF increased (especially at 150 mT), the effectiveness of the microalga and bacteria consortium in removing pollutants (e.g., phosphorus) from the reservoir increased.

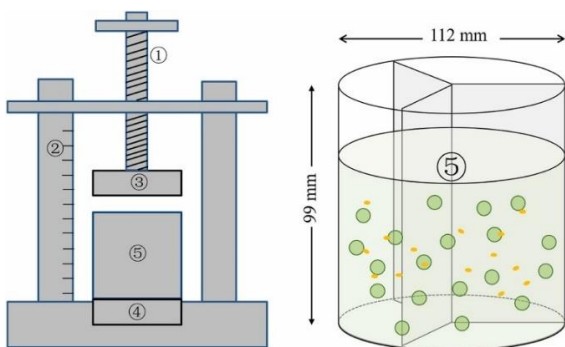

**Figure 5.** SMF stimulation device for the algal–bacterial consortium: 1—spiral adjustment rod, 2—scale plate, 3—movable magnetic pole, 4—permanent magnetic pole, 5—specially designed container. Source: [87].

Microalga growth in municipal wastewater converts nutrients into biomass containing valuable lipids and cleans the wastewater [5,87–90]. In a study presented by Feng et al. [91], two devices for wastewater treatment with microalga (*Chlorella pyrenoidosa*) were designed—with a bottom MF (magnetic induction in the range of 100–500 mT) and with a bypass MF (magnetic induction in the range of 50–500 mT)—Figure 6a,b, respectively. The upper MF pretreatment mode demonstrated a positive effect on the subsequent accumulation of biomass and lipid content. The optimal lipid production was obtained for the bottom MF at 500 mT for 1 h and the bypass MF at 500 mT for 3 h, respectively.

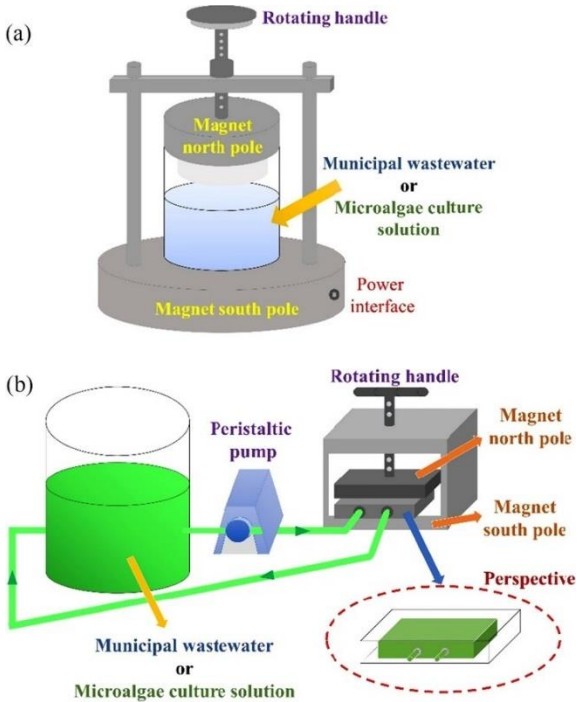

**Figure 6.** Diagram of devices for (**a**) low MF and (**b**) bypass MF. Source: [91].

A very similar concept of reactors may be found in many studies on wastewater treatment. Most of these concepts remain exclusive to the laboratory scale (indoors). An example of this was proposed by Tu et al. [4] and presented in Figure 7. It consists of magnets with a laminated iron core, a beaker, and a magnetic rotor. This system was developed to generate MFs from 50 to 500 mT. MF application of 100 mT for 0.5 h in the *Scenedesmus obliquus* logarithmic growth phase increased the chlorophyll content by 11.5%, by comparison with the control sample (biological material exposed only to the Earth's MF). In addition, the oxygen production rate increased by 24.6% through magnetic stimulation, by comparison with the control.

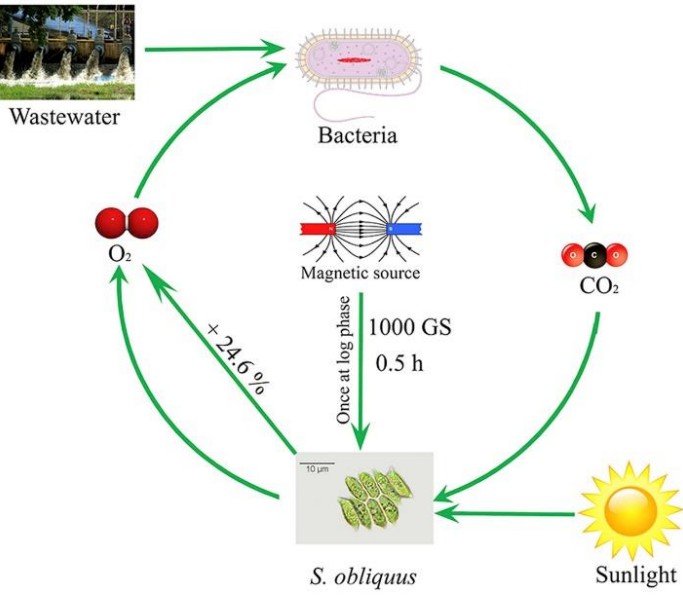

**Figure 7.** Static magnetic field (SMF) applied to *Scenedesmus obliquus* cultivated in municipal wastewater for oxygen production. Source: [4].

## 4. Sustainable and Environmentally Friendly Application of Microalgae

Microalgae present a high potential to produce many green chemicals, as already shown in this review, and because of that, these microorganisms may be treated as a "living biorefinery", considering that, while converting $CO_2$ and water to $O_2$, it is also possible to produce a rich biomass. Microalgae are represented as a sustainable and environmentally friendly raw material, since microalgal biomass may be obtained with low energy and costs with a good strategy. There are many ways to demonstrate that microalga biomass and cultivations may be applied as a sustainable and environmentally friendly utilization such as nutraceutics, food industry, and animal feed [3].

These microorganisms can grow with sunlight and use wastewater to "feed" themselves, such that this could be a more environmentally friendly way to obtain the biomass, since, as presented before, the microalgae can use the wastewater from different sources to replace the culture medium, besides performing wastewater treatment. In addition, microalgal biomass can be obtained year-round, unlike other crops, which can only be cultivated in particular periods [17]. This fact is important because we do not need to wait until the right season, nor do they require arable land, being an independent culture that would not affect the human food chain.

On the other side, to match the rising food demand, agrochemicals', such as pesticides/fertilizers, utilization on agricultural crops is currently increasing. Microalgae can also solve a part of this problem, because they can act as microbial biopesticides [92]. Biofertilizers can be used as nitrogen fixators, phosphates, and potassium-solubilizing biofertilizers, biofertilizers for secondary macronutrients such as iron and zinc, and phosphorus-mobilizing biofertilizers [93]. Microalgae have the ability to inhibit several pathogens

contaminating plant cultures [92,94] due to the different compounds (phenolic compounds and terpenes) present in the biomass, acting as growth regulators against pathogens [95,96].

The biofuel from microalga biomass is known as the third generation of biofuels that has been emerging in the world [63]. It is expected that this class of biofuels will minimize the dependency on fossil fuels, including the environmental issues related to their use, such as pollution and the increase in the greenhouse effect. Among the diverse sources for this generation of biofuels, microalga biomass presents a high potential to replace fossil fuels, being a renewable, nontoxic, and eco-friendly source. The oils present in microalga biomass have similar properties as vegetable oils. According to the Global Market Insight (GMI) report, alga-based biofuels may substitute petroleum-based fuels in emerging technological and economic sectors worldwide [97]. It is estimated by the U.S. Renewable Fuels Standard that approximately 36 billion gallons of microalga-based biofuels will have been produced in the year 2022 [98].

Regarding sustainable aquatic and terrestrial animal feed, microalgae also present a very important role. These microorganisms can be a good alternative because, according to Dineshbabu et al. [99], their biomass is nutritionally richer than the traditional ones with respect to the protein, carotenoids, omega 3, and fatty acids content; besides, they contain antioxidative, antimicrobial, and disease-preventing molecules.

Currently, the population is increasing exponentially, and the projection is for it to increase to 9.8 billion by 2050 [100]. According to Hunter et al. [101], food production should increase by 25–60% to feed the entire population. On the other hand, aquaculture involves related activities such as rearing, breeding, and harvesting of freshwater and marine species of aquatic plants and fish, and around 70% of aquaculture yield worldwide is produced using external feed [100]; besides, the aquaculture emission of greenhouse has been increasing [102]. Then, in this scenario, microalgae also play a role in being economical and efficient, because using the biomass as feed or as a feed supplement can reduce the fish meal based on aquaculture. In the same way, microalgae can be mixed with animal feed and decrease the requirements for grain, which can be used to feed people.

## 5. Final Considerations

This review presented the current trends about biostimulation in microalga cultivation, which is linked to biotechnology, good environmental protection, as well as bioenergy production, such as biofuels and biodiesels. Microalgae are a potential alternative sustainable source of energy, since biocompounds of great value can be obtained in an environmentally friendly way.

Microalgal-based biorefineries allow the sustainable generation of numerous high-value products from microalga biomass, allying biomass production with the use of natural resources. Besides, it is possible to integrate the circular bioeconomy to achieve the bioproducts' production, fix atmospheric $CO_2$, and valorize waste resources in biomass production.

Magnetic field application is a sustainable way to increase these factors and help apply the biorefinery concept. In the meantime, it is necessary to achieve the optimization of this process for each strain, even if this is a hard process, because each strain presents different behaviors under MF application, which is an area that needs to be explored more. On the other hand, it is already possible to find many studies in progress that show us the importance and representativity of this new field in microalgal biotechnology.

**Author Contributions:** Conceptualization, K.M.D. and M.S.; writing—original draft preparation, K.M.D., K.D., I.M., P.G.P.S., L.O.S., J.D. and M.S., writing—review and editing, S.K., M.I., M.B. and M.S.; visualization, K.M.D. and I.M.; supervision, M.I. and M.S. All authors have read and agreed to the published version of the manuscript.

**Funding:** This research received no external funding.

**Institutional Review Board Statement:** Not applicable.

**Informed Consent Statement:** Not applicable.

**Data Availability Statement:** Not applicable.

**Conflicts of Interest:** The authors declare no conflict of interest.

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
