# Peer review of "Microalgae Cultivated under Magnetic Field Action: Insights of an Environmentally Sustainable Approach"

_sustainability, doi:10.3390/su142013291_

Round 1
Reviewer 1 Report
General comments:
· Despite the fact that this review is well-organised and sets out interesting content regarding microalgae and their potential beneficial effects on the environment, it needs revision before it can be positively evaluated for publication.
-Lines 85-86:The authors stated: “The aim of this review is to present the current trends in electromagnetic bio-stimulation of microalgae for application of biomass in different areas of biotechnology, biofuel and bioenergy production, as well as environmental protection”. In order to improve the quality of the manuscript, I suggest the authors use these previous studies as references https://doi.org/10.1080/00207233.2021.1954775 and cite them in the text.
-Lines 295-310: Considering the information reported in this section, it would appear that the use of MF for the growth of certain species of bioalgae does not yield positive results in terms of benefits due to the triggering of oxidative stress processes, so I would like to ask the authors if they have done any research on this subject beyond that reported, and perhaps if they had thought to include information on comparative studies on the effects of MF-induced oxidative stress on various types of algae.
Author Response
Reviewer 1
Despite the fact that this review is well-organized and sets out interesting content regarding microalgae and their potential beneficial effects on the environment, it needs revision before it can be positively evaluated for publication.
Author’s response: Thank you for your contribution. We have considered all suggestions and the manuscript was all revised and corrected.
- Lines 85-86: The authors stated: “The aim of this review is to present the current trends in electromagnetic bio-stimulation of microalgae for application of biomass in different areas of biotechnology, biofuel and bioenergy production, as well as environmental protection”. In order to improve the quality of the manuscript, I suggest the authors use these previous studies as references https://doi.org/10.1080/00207233.2021.1954775 and cite them in the text.
Author’s response: Thank you. New information has been added to the introduction and supported by similar information and results provided by Spanò et. al. (2021) (Introduction, sections 3 and 4).
- Lines 295-310: Considering the information reported in this section, it would appear that the use of MF for the growth of certain species of bioalgae does not yield positive results in terms of benefits due to the triggering of oxidative stress processes, so I would like to ask the authors if they have done any research on this subject beyond that reported, and perhaps if they had thought to include information on comparative studies on the effects of MF-induced oxidative stress on various types of algae.
Author’s response: Thank you. Recently a review article was published elucidating the different effects on cellular metabolism, including null and inhibitory effects. (https://doi.org/10.1007/s11274-022-03398-y). Thus, the text has been modified and new information has been added (Section 2.3).
Reviewer 2 Report
1. The similarity with refs 39%. Encourage to reduce.
2. The title does not reflect as review article.
3. Component in abstract should contain. 1) Introduction 2) Obj 3) Method 4) Result 5) Conclusion 6) Recommendation. Make sure your abstract has included that 6 component.
4. Normally the element in keyword is from the title.
5. provide LR (Literature review) about MF in microalga cultivation.
6. better to add more discussion in this part. Need to discuss on details.
a. Provides the example line 487
b. Please include example of application (from line 496 to line 503)
c. Encourages to add figure to represent that data statistic
d. how algae contribute toward sustainability. add more LR

Author Response
Reviewer 2:
- The similarity with refs 39%. Encourage to reduce.
Author’s response: Thank you for your comment. The text was revised, and the similarity was reduced considerably.
- The title does not reflect as review article.
Author’s response: Thank you for your comment. The title was revised to reflect a review article.
- Component in abstract should contain. 1) Introduction 2) Obj 3) Method 4) Result 5) Conclusion 6) Recommendation. Make sure your abstract has included that 6 component.
Author’s response: Thank you for your comment. However, this manuscript is a review article addressing the state-of-the-art about insights on microalgae cultivated under magnetic fields as an environmentally and sustainable approach. Thus, the proposed division does not apply to this manuscript.
- Normally the element in keyword is from the title.
Author’s response: Thank you for your comment. However, in our understanding, the use of different words than those used in the title of the manuscript facilitates the indexing of the topic by search engines.
- Provide LR (Literature review) about MF in microalga cultivation.
Author’s response: Thank you for your suggestion. The text was revised and improved. Please, see the newly added information on Introduction.
- Better to add more discussion in this part (Topic 4). Need to discuss on details.
Author’s response: Thank you for your suggestion. We have considered all comments.
a. Provides the example line 487
Author’s response: New information has been added to the topic (Section 4).
b. Please include example of application (from line 496 to line 503)
Author’s response: New information has been added to the topic (Section 4).
c. Encourages to add figure to represent that data statistic
Author’s response: Thank you for your suggestion, however, it is not possible to represent the data in a figure, because there is not much different data that justify adding a figure in section 4.
d. How algae contribute toward sustainability. Add more LR.
Author’s response: Thank you for your comment. The text was revised and improved. Please, see the newly added information on Topic 4 (Section 4).